# Incidence and predictors of dehydration among preterm neonates admitted to the neonatal intensive care unit of Southern Ethiopia hospitals: A prospective follow-up study

**Mequanint Ayehu Akele** [1]*, **Dires Birhanu Mihretie**[1], **Mitsiwa Ruffo**[2], **Tamalew Alemie Tegegn**[1], **Migbar Sibhat Mekonnen**[1], **Abraham Dessie Gessesse**[3], **Abel Desalegn Demeke**[1], **Kassa Genetu Alem**[4]

1 Department of Nursing, College of Health Science and Medicine, Dilla University, Dilla, Ethiopia,
2 Department of pediatrics, College of Health Science and Medicine, Dilla University, Dilla, Ethiopia,
3 Department of Nursing, College of Health Science and Medicine, Woldeya University, Dilla, Ethiopia,
4 Department of Midwifery, College of Health Science and Medicine, Dilla University, Dilla, Ethiopia

* mequaninitayehu@gmail.com

## Abstract

### Background

Dehydration is one of the causes of preterm morbidity and mortality. Many efforts are being implemented to decrease the cause of preterm mortality, but there is a gap regarding dehydration in generating research-supported evidence. Therefore, this study aimed to investigate the incidence and predictors of dehydration among preterm neonates admitted to the neonatal intensive care units of Southern Ethiopia hospitals.

### Methodology

A prospective, institution-based follow-up study was conducted among 363 maternal-preterm neonatal pairs. Data were collected using a checklist and entered into Epidata version 4.4.2.1, and analyzed using STATA version 14. Bi-variable and multivariable regression were computed using the Cox regression model. A statistical significance was declared at a p-value of <0.05 in line with a 95% confidence interval (CI) and hazard ratios.

### Result

This study was conducted among 363 maternal-neonatal pairs. The incidence of dehydration was 41 per 1000 patient-days, with a total of 3546 patient-days observed. Maternal hypertension (AHR) = 2.95, CI: 1.56, 5.58), phototherapy (AHR = 2.71, 95% CI: 1.37, 5.39), nurse to patient ration 1:2 (AHR = 2.62, 95% CI: 1.5, 4.6), kept in incubator (AHR = 3.38, 95% CI: 1.77, 6.45), asphyxia (AHR = 1.58,

**Data availability statement:** The data is available at github repository. All the readers of this paper can access using this link https://github.com/M12-21/DHN-IN-PRETERM-NEONATES.git

**Funding:** The author(s) received no specific funding for this work.

**Competing interests:** The authors declared that there is no competing interest.

CI: 1.09–2.30) and absence of every four hour dehydration assessment (AHR = 1.62, CI: 1.42, 3.62) were predictors for incidence of dehydration among neonates admitted in NICU.

## Conclusion and recommendation

In this study, the incidence of dehydration among preterm neonates was high (41 per 1000 patient-days observation). The major associated factors were being under phototherapy, kept in an incubator, maternal hypertension, nurse-to-patient ratio, dehydration assessment practice, and asphyxia. So, establishing strategies on these identified predictor variables could be essential to reduce the suffering of preterm neonates' dehydration.

## Introduction

Dehydration is defined as an abnormal decrease in plasma volume, which implies water loss from both intracellular and extracellular spaces (intravascular and interstitial spaces). It can be occur broadly either from more fluid loss, reduced fluid intake, or both [1].

Newborn dehydration can be caused by neonatal infections as well as lactation problems related to first-time parenthood, maternal illnesses, inadequate lactation support for women, early postpartum discharge, and improper use of formula feeds [2]. These newborns may also have elevated bilirubin levels that require phototherapy, which exposes them to a water loss in the form of evaporation, combined with dehydration [3].

In addition, since preterm neonates have a smaller surface area, immature skin, and other organs crucial for maintaining body fluids, most of them are at risk for insensible water loss compared to term neonates [4,5]. Further, the majority of preterm infants lack the physiological development necessary to consume all of their essential fluids and nutrients through oral means. So, premature neonates are in the hands of their caregivers to control the amount of water they consume [6]. As a result, those neonates are exposed to a major change in total body fluid and electrolyte balance, which is not subject to homeostatic control [7]. Water loss through the skin and urine may exceed 200 mL/kg/day, which can represent up to one-third of total body water (TBW) [8]. Skin, mucous membranes, and the respiratory system account for two-thirds and one-third of all insensible water loss in preterm neonates, respectively [9]. This insensible water loss is higher in the first weeks of life. Consequently, the healthcare professional faces more problems to prevent detrimental oscillations in fluid and electrolyte balance and to maintain homeostasis during the transition from the fetal to the neonatal period, especially in the first week of life. As a result, preterm neonates develop dehydration soon after birth.

As evidence revealed, in the United States, 1.2 to 3.4 per 1000 live births developed dehydration, while in the Netherlands, its incidence was 58 per year per 100,000 live births [10,11]. In India, around 7% of neonates were affected by

dehydration. Further, the incidence of hypernatremia dehydration is estimated to be between 20 and 71 per 100,000 breast-fed infants [12]. In Africa, like Kenya, 16.7% of neonates were experiencing dehydration [13]. The burden of preterm dehydration, especially in developing countries, is worse due to clinical management gaps and lack of different medical equipment that are used to give IV fluid consistently, and absence of parenteral nutrition for sick babies. However, the true burden of preterm dehydration is not known in most developing countries like Ethiopia [14]. Most evidence indicates that disorders of fluid and electrolytes are common in neonates. As a result, a thorough understanding of the physiological changes in body water and solutes following birth is critical to ensuring a smooth transition from the aquatic in-utero environment [15].

Dehydration leads to different complications from seizures to intracranial thrombosis and is a potential fatal condition for neonates left untreated [15,16]. It may cause peripheral gangrene, convulsions, central venous and aortic thrombosis, coma and even death in the acute period; long-term neurodevelopmental abnormalities have also been reported [17,18].

Even if dehydration leads to different complications up to permanent organ damage and even to death, neither the World Health Organization nor country-based institutions working to improve neonatal health provides explicit recommendations for prevention, diagnosis, or management of neonatal dehydration. Moreover, the practice of dehydration diagnosis and treatment varies across the country based on the level of understanding and professional distribution [18]. As a result, the rate of fluid administration via infusion into the newborn's veins and arteries or by tube feeding into the stomach or intestine is decided by his doctors and nurses for the premature infant [19]. Consequently, most preterm neonates were mistreated. Therefore, determining the incidence and associated factors of dehydration among preterm neonates is essential to implement different interventions that reduce the suffering of preterm neonates with dehydration. So, the findings of this study will serve as baseline data to prepare common guidelines to diagnose and treat dehydration. Therefore, the aim of this study is to determine the incidence and associated factors of dehydration among preterm neonates.

## Method and materials

### Study area

This study was conducted in Gedeo Zone and Sidama Region selected public hospitals.

### Study design and period

An institution-based prospective follow-up study design was conducted from June 1, 2023, to May 30, 2024.

### Population

**Source of population.** All preterm neonates admitted to the NICU of Gedio Zone and Sidama Region Public Hospitals from June 1, 2023, to May 30, 2024.

**Study population.** All preterm neonates admitted to the NICU of Gedio Zone and Sidama Region selected Public Hospitals from **June 1, 2023, to May 30, 2024.**

### Inclusion and exclusion criteria

All eligible preterm neonates admitted and treated in neonatal intensive care units from June 1, 2023, to May 30, 2024, in Gedio Zone and Sidama Region Public Hospitals were included. Preterm neonates with major congenital anomalies, malnutrition, and referred from other health institutions with dehydration were excluded.

### Sample size determination and sampling procedure

**Sample size determination.** The sample size was determined by using a single population proportion formula by considering a p-value of 19.7% for the prevalence of a descriptive study conducted in Kenya [18], a margin of error of 5%,

and a confidence interval of 95% for the general objective. Based on this, the total sample size, by adding non-response rates, was 267. In addition, to get the maximum sample size, we determine the sample size by using Epi-Info version 7.7statistical software for the second objective by taking the odds ratio of the identified risk factors of dehydration, such as presence of nipple problems, caesarian delivery, and delays initiation of breastfeeding, maternal age, and neonatal age. Therefore, the neonatal age was given the maximum sample size (330). By adding a 10% non-response rate, the total sample size was 363, which is greater than that of the general objective. Therefore, 363 were decided to be the sample size of this study.

**Sampling technique and procedure.** Three public hospitals in the Gedeo zone and the Sidama region were selected by a lottery method. Three-month average baseline preterm admission data were taken from the HMIS (Health Management Information System) from each selected hospital. Based on the data, Dilla University Hospital has a total monthly preterm admission of 17, Yirgalem General Hospital has 20, and Hawasa University Comprehensive Specialized Hospital has an average monthly admission of 25, which yields a total of 62 monthly admissions in the Gedo Zone and Sidama region in selected public hospitals. Our calculated sample size is 363, and the annual estimated total preterm admissions were 62 × 12, which gives a total of 744 preterm neonates. From this, the value of k was calculated (total population/Sample size = 744/363) = 2.

Therefore, by using a systematic sampling technique, the first person was selected by lottery method from the two consecutive preterm admissions that fulfilled the inclusion criteria; then, every two people were interviewed and followed until the required sample size was achieved in each hospital. If the recruited sample index mother refused to participate, the next admission was recruited.

To determine how many subjects must be recruited from each study area, the calculated sample size was proportionally allocated to each hospital based on the average monthly flow of cases from previous month reports. By using a proportional allocation formula, a total sample of 100, 117, and 146 preterm neonates was recruited from Dilla, Yirgalem, and Hawasa referral hospitals.

## Operational definition

**Dehydration (event):** In this study, dehydration stands for preterm neonates with weight loss (a weight loss (≥20%) in the first week of life (current weight minus birth weight) and ≥10% after the first week of life) combined with having two/more any of the following: having signs of oral dry mucosa, depressed fontanel plus increased serum sodium ($Na > 145$ mEq/L), increased urine specific gravity (>1.020), urine osmolality (>400 mosm/L), or decreased urine output (<1 ml/kg/hr) were considered as dehydration after admission. This definition was settled by combining ideas from physician experience and case definition of studies used in Netherlands, California, and Kenya [18, 20].

**Censored:** Preterm neonates, who didn't develop dehydration during follow-up, including transfer out, death, or loss to follow-up without dehydration, didn't develop dehydration until the end of follow-up.

**Preterm**: neonates who are diagnosed as preterm either by last normal menstrual period, by Ballard score, or using early ultrasound (20 weeks).

**The entry date** was the date in which the preterm neonates were admitted.

**The end date** was the last date in which each observation within the study period was last observed.

## Method of data collection and data collection tool

Data collection tool was adapted by reviewing related literature [18,20–22] and modified according to our context. The data was collected by using the checklist that includes socio-demographic characteristics of the mother, the maternal obstetric problems during the pregnancy, gestational age, age of the neonate, and clinical co-morbidities of the neonate during admission and follow-up.

## Data quality assurance

Two days of training and clear orientation were given for all data collectors by principal investigators (6 BSc neonatal nurses) and supervisors (3 MSc neonatal nurses) on how and what information they should collect and ethical considerations for collecting the data. A pretest was employed on 5% (n = 22) of the sample size by using the data collection tool in Adare General Hospital. In addition, the close supervision of data collectors by the principal investigator was also continued during the progress of data collection.

## Data analysis

The collected data were coded and entered into EpiData version 4.4.2.1 by the principal investigator. The entered data were exported and analyzed using STATA version 14. Descriptive statistics like frequency, proportions, measures of central tendency, and standard deviation were computed to describe the study variable in relation to the population. Multicollinearity among independent variables was checked by using a variance inflation factor (VIF) (1.43). The Kaplan-Meier curve was constructed to determine the time to dehydration occurrence and cumulative survival probabilities at a point in time. The long-rank test was used to compare the survival status between categorical variables. In addition, the Cox PH regression model was applied to conduct bivariable and multivariable analysis. Variables with $p \leq 0.25$ in the bivariable analysis were transferred to multivariable analysis for confounder adjustment. PH assumptions were checked using the Schoenfeld residuals test (global test = 0.17), and the Cox-Snell residual test was used to test the model's fitness. The findings showed that the PH assumptions were fulfilled, and the model was fitted well to the data. Finally, statistical significance was declared at a $p$-value of <0.05 in line with a 95% confidence interval (CI). The hazard ratio was used to report the strength of associations.

## Ethical statement

This study was conducted according to the Declaration of Helsinki. Ethical approval was obtained from the Institutional Review Board of Dilla University, College of Health Sciences and Medicine (protocol unique No.: duirb/041/23-06). Written informed consent was obtained from the guardians of the children to participate in the study before the study began.

## Result

### Maternal and neonatal socio-demographic characteristics

This study included a total of 363 mother-neonate cohorts with a 100% response rate. Among those study participants, two hundred ninety-two (80.5%) of mothers were found in the age group of 21–35 years old. Two hundred four (56.2%) of mothers reside in urban settings. Among these participants, 50.1% of the neonates were females. In addition, 81.5% of the preterm neonates were born at a gestational age of greater than 32 weeks. Further, 67.2% of the preterm neonates' weights are between 1500 and 2500 grams. The mean admission time of the preterm neonate to the NICU was 4 hours after delivery. When compared with neonates admitted within 1 hour and later, 41 (11.3% and 107 (29.5%) preterm neonates developed dehydration among 363 preterm neonates, respectively (Table 1).

### Maternal medical and obstetrics characteristics

One hundred seventy-seven (48.7%) of mothers had ≥3 antenatal care (ANC) follow-ups. Regarding the mode of delivery, 49% of mothers were primigravida, and 56.5% gave birth by spontaneous vaginal delivery. In addition, a total of one hundred thirty-five (37.2%) mothers had developed a breast problem after delivery. Among 363 participants, 241 (66.4%) mothers had at least one risk factor for preterm birth. Regarding chronic disease, 23.7% had a history of chronic disease, such as anemia, 9.4% and 8.8% diabetes mellitus (Table 2).

**Table 1. Socio-demographic characteristics of mothers and neonates' admitted at Gedeo zone and Sidama region public hospitals in 2023/2024 (n = 363).**

| Variables | Categories | Frequency | Percentage |
|---|---|---|---|
| Age | <23 | 27 | 7.4 |
| | 23–35 | 292 | 80.5 |
| | >35 | 44 | 12.1 |
| Residence | Urban | 204 | 56.2 |
| | Rural | 159 | 43.8 |
| Educational status | Illiterate | 62 | 17.1 |
| | Primary school | 133 | 36.6 |
| | Secondary school | 89 | 24.5 |
| | Diploma | 54 | 14.9 |
| | Degree and above | 25 | 6.9 |
| Marital status | Married | 287 | 79 |
| | Single | 32 | 8.8 |
| | Divorced | 22 | 6.1 |
| | Windowed | 22 | 6.1 |
| Sex of the neonate | Male | 181 | 49.9 |
| | Female | 182 | 50.1 |
| Age at admission in hr | Mean ±SD | 4 ± .442 | – |
| Gestational age in w/k | ≤ 32 w/k | 67 | 18.5 |
| | > 32w/k | 296 | 81.5 |
| Birth weight in gram | < 1500 | 96 | 26.5 |
| | 1500–2500 | 244 | 67.2 |
| | ≥ 2500 | 23 | 6.3 |
| Weight for gestational age | AGA | 239 | 65.8 |
| | LGA | 22 | 6.1 |
| | SGA | 102 | 28.1 |

### Neonatal medical problem

More than half (55.1%) of the neonates had a first one-minute Apgar score found to be in the range of 7–10, and 82.4% had a first five-minute score of 7–10. From a total of study neonates, 44.4% were hypothermic, followed by respiratory distress (30.6%) during admission, and 41.9% experienced a new medical problem during follow-up. Of those who developed new medical problems, 18.2% had anemia and 13.2% had hospital-acquired infections (Table 3).

### Treatment and health service related characteristics

Regarding treatment and health service-related problems, half of the neonates (50.7%) were kept in radiant warmers, and 13.2% were under phototherapy. One hundred eight (29.8%) were treated with CPAP, and 36.4% of neonates didn't take anything per month. More than half (58.7%) of the nurses deliver care for admitted neonates with a 1:2 ratio, and 46.6% of the neonates were assessed every four hours for dehydration (Table 4).

### Incidence of dehydration in preterm neonates admitted to the NICU

A total of 363 preterm neonates were followed for a minimum of 1day and a maximum of 28 days. Among those neonates, 41% were developing dehydration. The incidence of dehydration was 41 per 1000 patient-days of observation, with a total follow-up time of 3546 preterm days of observation. The cumulative incidence rate of dehydration in the first seven days,

**Table 2. Shows maternal medical and obstetrics characteristics of mothers at Gedeo Zone and Sidam region public hospitals in 2022/2024 (n = 363).**

| Variables | Categories | Frequency | Percentage |
|---|---|---|---|
| ANC follow up | Yes | 293 | 80.7 |
| | No | 70 | 19.3 |
| Number of ANC follow up | 0 | 70 | 19.3 |
| | <3 | 116 | 32 |
| | ≥3 | 177 | 48.7 |
| Gravidity | Primigravida | 178 | 49 |
| | Multigravida | 185 | 51 |
| Type of pregnancy | Singleton | 306 | 84.3 |
| | Multiple | 57 | 15.7 |
| Mode of delivery | SVD | 205 | 56.5 |
| | Instrumental | 72 | 19.8 |
| | C/S | 86 | 23.7 |
| Breast problem | Yes | 135 | 37.2 |
| | No | 228 | 62.8 |
| Risk of preterm birth | Yes | 241 | 66.4 |
| | No | 122 | 33.6 |
| PROM | Yes | 76 | 20.9 |
| | No | 287 | 79.1 |
| APH | Yes | 81 | 22.3 |
| | No | 282 | 77.7 |
| Pre-eclampsia | Yes | 31 | 8.5 |
| | No | 232 | 91.5 |
| Eclampsia | Yes | 47 | 13 |
| | No | 316 | 87 |
| Oligo-hydroaminous | Yes | 38 | 10.5 |
| | No | 325 | 89.5 |
| Poly-hydroaminous | Yes | 52 | 14.3 |
| | No | 311 | 85.7 |
| Infection | Yes | 41 | 11.3 |
| | No | 322 | 88.7 |
| Chronic disease | Yes | 86 | 23.7 |
| | No | 277 | 76.3 |
| DM | Yes | 32 | 8.8 |
| | No | 331 | 91.2 |
| HPT | Yes | 25 | 6.9 |
| | No | 338 | 93.1 |
| Anemia | Yes | 34 | 9.4 |
| | No | 329 | 90.6 |
| HIV AIDS | Yes | 24 | 6.6 |
| | No | 339 | 93.4 |

between seven days and fourteen days, and after fourteen days was 69, 14.9, and 7.2 per 1000 patient-days of observation, with a total of 3546 patient-days of observation, respectively. The probability of the occurrence of dehydration at 7 days, 14 days, and 28 days using the Kaplan-Meier list was 36%, 42%, and 72% respectively.

**Table 3. Shows neonatal medical problems during admission and follow-up at Gedeo Zone and Sidama region public hospitals, 2022/2024 (n = 363).**

| Variables | Categories | Frequency | Percentage |
|---|---|---|---|
| The first one minute APGAR score | < 4 | 5 | 1.4 |
| | 4–7 | 158 | 43.5 |
| | 7–10 | 200 | 55.1 |
| The first five minute APGAR score | < 4 | 1 | 0.3 |
| | 4–7 | 63 | 17.3 |
| | 7–10 | 299 | 82.4 |
| **Diagnosis at admission** | | | |
| Hypothermia | Yes | 161 | 44.4 |
| | No | 202 | 55.6 |
| Respiratory distress | Yes | 111 | 30.6 |
| | No | 252 | 69.4 |
| Hypoglycemia | Yes | 81 | 22.3 |
| | No | 282 | 77.7 |
| Infection | Yes | 38 | 10.5 |
| | No | 325 | 89.5 |
| Asphyxia | Yes | 121 | 33.3 |
| | No | 242 | 66.7 |
| Jaundice | Yes | 36 | 9.9 |
| | No | 327 | 90.1 |
| Others | | | |
| **New medical problem during follow up** | | | |
| Presence of new medical during follow up | Yes | 152 | 41.9 |
| | No | 211 | 58.1 |
| Hospital acquired infections | Yes | 50 | 13.8 |
| | No | 313 | 86.2 |
| Jaundice | Yes | 41 | 11.3 |
| | No | 322 | 88.7 |
| Respiratory distress syndrome | Yes | 23 | 6.3 |
| | No | 340 | 93.7 |
| Thrombocytopenia | Yes | 28 | 7.7 |
| | No | 335 | 92.3 |
| Anemia | Yes | 66 | 18.2 |
| | No | 297 | 81.8 |

## KM curve

As Fig 1 notes, the probability of developing dehydration increases sharply in the first weeks and rises stepwise to 40%. In addition, the graph crosses the 50% failure probability at 28 days. This indicates that 50% of the participants will have a probability of developing dehydration after 28 days of observation.

## PH assumption test and model fitness

The Schoenfeld residual test was used to test the Cox PH assumption for each individual variable. The overall global test of all covariates was 0.17, which is above the P-value of 0.05. It was noted that none of the variables violated the PH assumption test, and the model fitted the data well.

**Table 4. Shows treatment and health service-related characteristics of the preterm neonates admitted in Sidama region and Gedeo Zone public hospitals in 2022/2024 (n = 363).**

| Variables | Categories | Frequency | Percentage |
|---|---|---|---|
| Dehydration | Yes | 148 | 40.8 |
| | No | 215 | 59.2 |
| Date of dehydration | | | |
| Neonates kept | Regular bed (without radiate warmer, incubator and phototherapy) | 84 | 23.1 |
| | Radiate warmer | 184 | 50.7 |
| | Incubator | 47 | 13 |
| | Phototherapy | 48 | 13.2 |
| CPAP | Yes | 108 | 29.8 |
| | No | 255 | 70.2 |
| Does the neonate kept NPO | Yes | 132 | 36.4 |
| | No | 231 | 63.6 |
| Is the neonate on Iv fluid | Yes | 178 | 49 |
| | No | 185 | 51 |
| If yes, amount of fluid in ml | Mean ±SD | 296.3 ± 137.7 | |
| Nurse to patient ratio | 1:1 | 73 | 20.1 |
| | 1:2 | 213 | 58.7 |
| | > 1:2 | 77 | 21.2 |
| Dehydration assessment every four hr | Yes | 169 | 46.6 |
| | No | 194 | 53.4 |

## Predictors of dehydration

In bivariate Cox regression, 14 variables were candidates for multivariable Cox regression analysis. From those variables, 6 variables such as maternal hypertension, phototherapy, keeping in an incubator, nurse-to-patient ratio, asphyxia, and absence of every four-hour dehydration assessment were predictors for the incidence of dehydration among neonates admitted to the NICU.

Accordingly, the hazards of preterm neonates developing dehydration were 2.95 times higher among mothers who lived with hypertension (adjusted hazard ratio (AHR) = 2.95, CI: 1.56, 5.58) as compared to mothers without hypertension. Further, preterm neonates under phototherapy were 2.71 times more at increased hazard of dehydration (AHR = 2.71, 95% CI: 1.37, 5.39) as compared to preterm neonates kept in bed. Moreover, preterm neonates in incubators increase the hazard of dehydration by 3.38 times as compared to preterm neonates kept in bed (AHR = 3.38, 95% CI: 1.77, 6.45). Likewise, preterm neonates who didn't have a dehydration assessment every four hours had a 1.6 times increased hazard of developing dehydration as compared to their counterparts. In addition, a nurse-to-patient ratio of 1:2 increases the hazard of dehydration 2.62 times (AHR = 2.62, 95% CI: 1.5, 4.6) as compared to a nurse-to-patient ratio of 1:1. Similarly, nurse-to-patient ratios greater than 1:2 were 2.92 times more likely to increase the hazard of dehydration (AHR = 2.92, 95% CI: 1.48, 5.71) as compared to nurse-to-patient ratios of 1:1. A preterm neonate with asphyxia increases the hazard of developing dehydration by 1.63 times (AHR = 1.58, C: 1.09–2.30) as compared to a preterm neonate without asphyxia (Table 5).

## Discussion

The finding of this study focuses on the incidence and predictors of dehydration among neonates admitted to the neonatal intensive care unit. So, the findings of this study are essential to prevent the death and suffering of preterm neonates

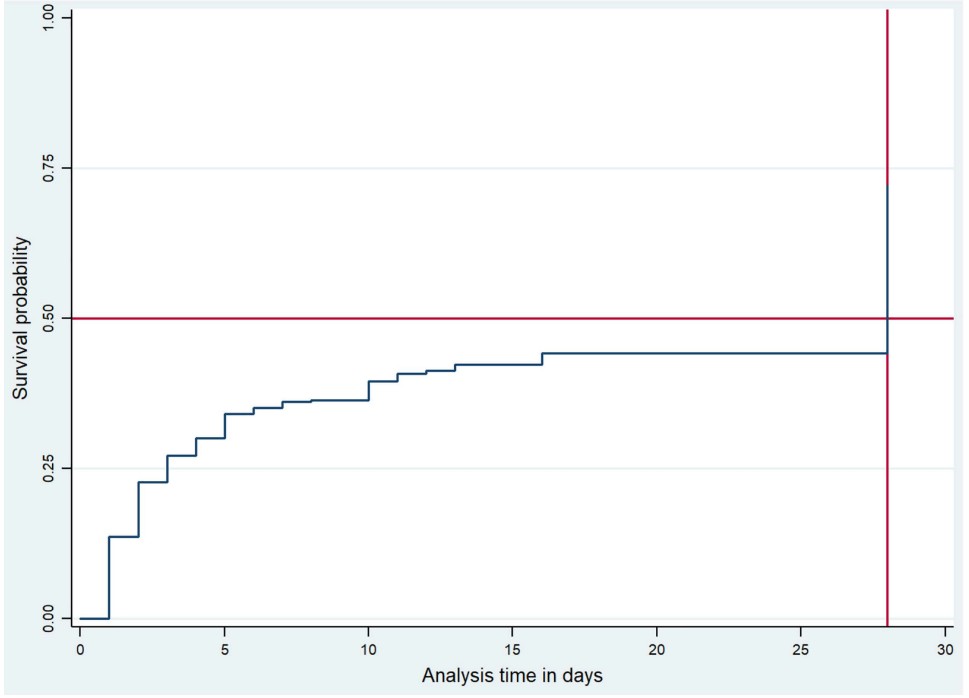

**Fig 1. Showed probability of developing dehydration among preterm neonates admitted in the Sidama region and Gedeo Zone public hospitals in 2022/2024.** The x-axis showed the analysis time in day and the y-axis showed the survival probability of dehydrated neonates in percent. In addition, the red line in the graph is serve as reference lines that indicate 50% of neonates develop dehydration at 28 days.

with dehydration and its complications by directly acting on the predictors that are associated with the occurrence of dehydration.

According to this study's findings, the incidence of dehydration was 41 per 1000 patient-days of observation. This finding is higher than studies conducted in the Netherlands and northern California (39.7 per 1000 and 2.1 per 1000) [23,24]. The possible justification for this variation could be a difference in population, in measurement, and in the case definition of dehydration used to diagnose dehydration for the study participants in both studies conducted in the Neatherland and California [11,20]. In addition, this variation could be a difference in the quality of care, availability of care, accessibility of standardized materials important for diagnosis and monitoring, and treatment options to prevent dehydration. In addition, the difference in sample size and the study population (6834, term and preterm in the first two weeks) in the case of the Netherlands may be the possible difference for variation in incidence of dehydration [10].

This study also noted that maternal hypertension is significantly associated with the incidence of dehydration in neonates. The scientific explanation for this association could be the presence of hypertension in pregnancy can lead to reduced blood flow to the placenta, which in turn restricts the fetus's access to nutrients and oxygen that potentially cause the baby to be born with lower fluid reserves, leading to dehydration after birth [25]. In addition, mothers with hypertension during pregnancy are at high risk of preterm birth. Preterm babies had immature kidney function, which made them more susceptible to dehydration [26]. Moreover, women who were hypertensive during pregnancy had insufficient breast milk supply. Consequently, the preterm neonate may get in adequate fluid. As result, the neonates may become dehydrated. This concept is supported by study conducted in Canada [23]. In the same fashion, putting of preterm neonates under phototherapy was significantly associated with incidence of dehydration in preterm neonates. It is supported by evidence revealed in a study conducted in India [24]. The scientific explanation for this association could be due to the warming

**Table 5. Shows multivariable Cox proportional hazard regression analysis for the predictors of dehydration among preterm neonates admitted in the Sidama region and Gedeo Zone public hospitals in 2022/2024.**

| Variable | Category | Dehydration diagnosis | | p-value | Adjusted hazard ratio (95% CI) |
|---|---|---|---|---|---|
| | | Censored | Events | | |
| Educational status | Illiterate | 26 | 36 | 0.27 | 1.65 (0.67, 4.06) |
| | Primary school | 55 | 78 | 0.41 | 1.43 (0.61, 3.39) |
| | Secondary school | 37 | 52 | 0.3 | 1.62 (0.66, 3.97) |
| | Diploma | 23 | 31 | 0.49 | 1.39 (0.55, 3.49) |
| | Degree and above | 7 | 18 | 1 | 1 |
| Marital status | Married | 121 | 126 | 1 | 1 |
| | Single | 8 | 24 | 0.15 | 0.57 (0.26, 1.23) |
| | Divorced | 7 | 15 | 0.27 | 1.6 (0.68, 3.77) |
| | Windowed | 12 | 10 | 0.56 | 0.79 (0.36, 1.73) |
| Mode of delivery | SVD | 91 | 114 | 1 | |
| | Instrumental | 22 | 50 | 0.36 | 0.7 (0.45, 1.33) |
| | C/S | 35 | 51 | 0.25 | 1.29 (0.83, 2.02) |
| Maternal hypertension | Yes | 17 | 8 | 0.001 | 2.95 (1.56, 5.58)* |
| | No | 131 | 207 | | |
| Maternal diabetes mellitus | Yes | 9 | 23 | 0.28 | 0.66 (0.32,1.39) |
| | No | 139 | 192 | | |
| Jaundice during admission | Yes | 21 | 15 | 0.19 | 1.49 (0.82, 2.69) |
| | No | 127 | 200 | 1 | |
| Where the neonate Kept | Bed | 22 | 62 | | |
| | Radiate warmer | 55 | 129 | 0.80 | 1.08 (0.61,1.91) |
| | Incubator | 42 | 5 | 0.001 | 3.38 (1.77, 6.45)* |
| | Phototherapy | 29 | 19 | 0.004 | 2.71 (1.37,5.39)* |
| DHN assessment practice | Yes | 54 | 73 | 1 | |
| | No | 94 | 142 | 0.001 | 1.62 (1.42, 3.62)* |
| Nurse to patient ratio | 1:1 | 20 | 53 | 1 | |
| | 1:2 | 93 | 120 | 0.001 | 2.62 (1.5, 4.6)* |
| | > 1:2 | 35 | 42 | 0.002 | 2.92 (1.48, 5.71)* |
| Age of the mother | 1 | 7 | 20 | 0.63 | 0.81 (0.32, 1.93) |
| | 2 | 122 | 170 | 1 | |
| | 3 | 19 | 25 | 0.29 | 0.72 (0.39, 1.32) |
| Type of feeding | Trophic feeding | 77 | 80 | 0.95 | 1.01 (0.65, 1.57) |
| | Complete breast feeding | 54 | 126 | 1 | |
| | NPO | 17 | 9 | 0.09 | 1.77 (0.92, 3.41) |
| RDS during follow up | Yes | 6 | 21 | 0.06 | 0.37 (0.13, 1.06) |
| | No | 144 | 196 | | |
| Asphaxia | Yes | 66 | 55 | 0.016 | 1.58 (1.09 2.30)* |
| | No | 82 | 160 | | |
| Birth weight | <1500 | 51 | 45 | 0.52 | 1.44 (0.49, 4.43) |
| | 1500-2500 | 145 | 99 | 0.54 | 1.4 (0.48, 4.04) |
| | >2500 | 19 | 4 | 1 | |

*variables with p-value less than 0.05.

effect of phototherapy, increasing insensible water loss through evaporation of the skin. Phototherapy also potentially causes diarrhea that leads to loss of fluid from the body. As a result, the neonates develop dehydration. Similarly, keeping preterm neonates in an incubator increases the hazard of developing dehydration in preterm neonates. Even if the incubators are designed to maintain a stable environment for premature infants, providing warmth and regulating humidity, incubators lead to the loss of so much water through sweating or increasing respiratory rate, especially if the temperature in the incubator is not carefully monitored or adjusted, since the skin of the neonates is immature. Consequently, those neonates develop dehydration. Moreover, the absence of every four-hour dehydration assessment was a predictor of dehydration. The possible explanation for this association could be the absence of dehydration assessment practice may lead to missing of early prevention of factors that lead to dehydration. As a result, the neonate may develop dehydration.

Nurse-to-patient ratio is one of the predictors of dehydration in preterm neonates. An equal proportion of nurses to patients are necessary for strict follow-up of the patient's condition, especially for preterm neonates admitted to the neonatal intensive care unit since the preterm neonates are at risk for different types of life-threatening conditions. Consequently, if the ratio of nurses to patients is not one to one, the contacts of nurses with the neonates become decreased. As a result, preterm neonates may develop dehydration. Likewise, preterm neonates who had asphyxia were more likely to develop dehydration as compared to preterm neonates without asphyxia. The scientific explanation for this association could be due to asphyxia: the body of the neonates' experiences stress that triggers the production of antidiuretic hormone that causes retention of fluid in the body. This accumulated or retented fluid leads to dilution of blood. This causes dilutional hyponatremia. As a result, the neonates become dehydrated. In addition, neonatal asphyxia induces blood flow centralization. During asphyxia, there is intense sympathetic activation and a rise in catecholamines. Due to this, cardiac output is redistributed to vital organs (brain, heart, adrenal glands), with peripheral vasoconstriction and reduced renal and splanchnic blood flow. This renal hypoperfusion impairs the ability to concentrate urine and retain sodium and water, contributing to increased urinary losses during the post-reperfusion phase [27,28]. Furthermore, asphyxia leads to hypoxic–ischemic tubular injury. As a result, urinary loss of sodium (natriuresis) and chloride increases, in addition to impaired tubular water reabsorption, leading to fluid deficiency [29]. Combined with hormonal dysregulation and increased insensible losses, these changes predispose affected neonates to more rapid and severe dehydration compared to those without asphyxia [30,31].

## Conclusion

In this study, the incidence of dehydration in preterm neonates is higher than previously reported in studies at the national and international levels. Maternal hypertension, phototherapy, keeping the infant in an incubator, asphyxia, absence of every four-hour dehydration assessment practice, and nurse-to-patient ratio were identified as independent predictors of dehydration of preterm neonates. So, setting and designing new strategies and strengthening the existing strategies, focusing on the identified predictors of dehydration, is essential to decrease the suffering of preterm neonates with dehydration and its complications.

## Strengths and limitations

This study's finding contributes essential points relevant for improving neonatal care, since the study specifically addresses the high-risk groups, preterm neonates, who are more susceptible to dehydration due to their undeveloped physiological system. In addition, this study considers censored data and time variables in the analysis, and a relatively large sample size increases the validity of the findings. However, this study didn't consider all potential predictors of dehydration, such as genetic factors, limiting the scope of the findings. In addition, the criteria used for the diagnosis of dehydration were so broad. This may inflate the estimate of the incidence of dehydration. On the other hand, a number of studies used 10–15% weight loss to diagnose neonatal dehydration. However, in this study, a higher cut-off for weight loss (15% for neonates beyond 1 week and 20% for neonates below 1 week) was used to diagnose dehydration. So, this

may result in an underestimation of the incidence of dehydration. Further, the finding of this study was compared with the finding of previous study with different metrics because of the absence of available evidence directly conducted on preterm dehydration. On the other hand, as different literature stated, neonates with low gestational age were at risk for dehydration. However, in this study, gestational age was not included in the multivariable analysis. But by considering its clinical importance, we tried to include it in the multivariable analysis. However, while we tested model fitness, it also broke the model assumptions. Due to this, we didn't include the gestational age in the multivariable analysis. So, future researchers should take this gap and create their own hypothesis to prove it.

## Acknowledgments

First of all, the authors would like to thank Dilla University for giving them the opportunity to conduct this study. Next to this, we would like to extend our deepest gratitude to all the data collectors whose dedication, perseverance, and attention to detail made this work possible. Their commitment to ensuring the accuracy and integrity of the data, often in challenging conditions, was invaluable to the success of this study.

## Author contributions

**Conceptualization:** Dires Birhanu Mihretie, Mitsiwa Ruffo, Abraham Dessie Gessesse.

**Data curation:** Mequanint Ayehu Akele, Migbar Sibhat Mekonnen, Mitsiwa Ruffo, Abel Desalegn Demeke.

**Formal analysis:** Mequanint Ayehu Akele, Dires Birhanu Mihretie, Migbar Sibhat Mekonnen, Mitsiwa Ruffo, Abraham Dessie Gessesse.

**Funding acquisition:** Mequanint Ayehu Akele, Migbar Sibhat Mekonnen, Tamalew Alemie Tegegn, Kassa Genetu Alem.

**Investigation:** Mequanint Ayehu Akele, Dires Birhanu Mihretie, Migbar Sibhat Mekonnen.

**Methodology:** Mequanint Ayehu Akele, Dires Birhanu Mihretie, Migbar Sibhat Mekonnen, Tamalew Alemie Tegegn, Kassa Genetu Alem.

**Project administration:** Mequanint Ayehu Akele, Tamalew Alemie Tegegn, Kassa Genetu Alem.

**Resources:** Mequanint Ayehu Akele, Mitsiwa Ruffo, Abel Desalegn Demeke.

**Software:** Mequanint Ayehu Akele, Dires Birhanu Mihretie, Migbar Sibhat Mekonnen, Mitsiwa Ruffo, Tamalew Alemie Tegegn, Abraham Dessie Gessesse, Abel Desalegn Demeke.

**Supervision:** Mequanint Ayehu Akele, Kassa Genetu Alem, Abraham Dessie Gessesse, Abel Desalegn Demeke.

**Validation:** Mequanint Ayehu Akele, Migbar Sibhat Mekonnen, Mitsiwa Ruffo, Abel Desalegn Demeke.

**Visualization:** Mequanint Ayehu Akele, Dires Birhanu Mihretie, Mitsiwa Ruffo, Abraham Dessie Gessesse.

**Writing – original draft:** Mequanint Ayehu Akele, Dires Birhanu Mihretie, Kassa Genetu Alem.

**Writing – review & editing:** Mequanint Ayehu Akele, Dires Birhanu Mihretie, Migbar Sibhat Mekonnen, Abraham Dessie Gessesse.

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
