## [Decision Letter · Decision Letter 0]

26 Jun 2025

Dear Dr. Ayehu,

Thank you for submitting your manuscript to PLOS ONE. After careful consideration, we feel that it has merit but does not fully meet PLOS ONE’s publication criteria as it currently stands. Therefore, we invite you to submit a revised version of the manuscript that addresses the points raised during the review process.

We look forward to receiving your revised manuscript.

Kind regards,

Ricardo Q. Gurgel, PhD

Academic Editor

PLOS ONE

Journal Requirements:

2. In the online submission form, you indicated that the data underlying the results present in the study are available from reasonable request of the corresponding author.

Reviewers' comments:

Reviewer's Responses to Questions

**Comments to the Author**

1. Is the manuscript technically sound, and do the data support the conclusions?

Reviewer #1: Partly

Reviewer #2: Yes

2. Has the statistical analysis been performed appropriately and rigorously?

Reviewer #1: Yes

Reviewer #2: Yes

3. Have the authors made all data underlying the findings in their manuscript fully available?

Reviewer #1: Yes

Reviewer #2: Yes

4. Is the manuscript presented in an intelligible fashion and written in standard English?

Reviewer #1: No

Reviewer #2: Yes

Reviewer #1: Reviewer Comments

First, I would like to commend the authors for selecting the topic of dehydration in preterm neonates admitted to a NICU. This is an important subject due to its high frequency and association with significant morbidity and mortality, which are even more pronounced in developing countries. The topic is sparsely addressed in the literature and clearly required considerable effort from the research team in both data collection and interpretation.

I have read the manuscript thoroughly and identified certain inconsistencies and points of uncertainty, which I have listed below:

1. A critical concern relates to the definition of dehydration used for data collection, as it serves as the foundation for the entire statistical analysis. The methodology does not clearly define “dehydration” in a detailed manner. It appears from the text that dehydration was defined as a 20% loss of birth weight in the first week and a 10% loss after the first week, but other clinical signs and criteria associated with dehydration are mentioned, making it unclear how these were used in conjunction with weight loss. It would be ideal if the authors clearly stated the definition, for example: “dehydration was defined as a weight loss combined with at least one of the following clinical signs”; or “dehydration was defined as weight loss and all clinical signs listed”; or “dehydration was defined as weight loss and two or more clinical signs,” etc. Additionally, I would like to review the data collection form used, as this would help clarify the approach. This point needs to be rewritten to eliminate ambiguity, as it critically affects the rest of the analysis.

2. I recommend including a reference for the definition of dehydration in preterm infants, if available. The text should state clearly whether the definition was based on a prior published article, another author’s experience (with a cited source), or created exclusively for this study.

3. Another point for review is the measurement unit for dehydration. When comparing rates across studies, it is crucial to ensure that comparable units are used. In the literature I reviewed — including articles cited by the authors — dehydration incidence is highly variable due to differing definitions and objectives (e.g., dehydration in exclusively breastfed infants versus dehydration due to excessive fluid loss). Units of measurement also vary (percentage of dehydrated patients, dehydrated patients per 1,000 live births, dehydrated patients per 1,000 exclusively breastfed neonates, etc.). I could not find any publication using the same measurement (dehydration per 1,000 patient days) as used in this study. This discrepancy complicates direct comparisons with existing data. Additionally, inconsistencies arise in the text when the authors compare their results with the literature using differing units. For example, lines 238–239 seem to imply that an incidence of 41 per 1,000 patient days is equivalent to 41%, which is incorrect.

More generally, when the authors cite literature about dehydration definitions, two key studies are mentioned, yet reference 11 is not cited in the section dealing with definitions and comparisons. Upon review, I found that:

• In the Netherlands study (Reference 11), dehydration was defined as weight loss >10% of birth weight combined with a serum sodium ≥150 mEq/L. To minimize bias related to low intake and weight loss due to infection, neonates with UTIs and GERD were excluded.

• In the California study (Reference 22), dehydration was defined as either: (a) a serum sodium ≥150 mEq/L, and/or (b) weight loss ≥12%.

In lines 281–282, I could not locate the cited California reference (22), and the Netherlands reference is not mentioned (although I assume it is Reference 11). The Dutch article uses dehydration incidence per 100,000 exclusively breastfed neonates, while the Californian reference uses an incidence of 2.1 per 1,000 live births. These differing units must be clearly explained in the text, making it evident that the objectives and patient populations differ from those of the present study (dehydration in preterm neonates in a NICU). Additionally, it would be helpful to compare the current results with those from other studies using the same metric (dehydration per 1,000 patient days).

4. The results section contains inconsistencies between data presented in tables and those stated in the conclusion. For example, on line 266, an odds ratio (OR) of 3.38 suggests that incubators are a risk factor for dehydration. However, in the abstract, an OR of 0.19 is stated, implying a protective effect. This discrepancy must be resolved.

Moreover, upon closer inspection of the tables, it is unclear how bed types were classified. The text mentions incubators, radiant warmers, common cribs, and phototherapy units. However, it is ambiguous whether “common cribs” means a regular hospital crib or a bed adjacent to the mother. Additionally, why was phototherapy treated as a bed category, given that it can be administered in any of these environments? This may introduce confounding, as dehydration attributed to an incubator might, in fact, relate to phototherapy exposure.

Importantly, the results and tables do not assess gestational age as a potential risk factor for dehydration. According to literature, extremely preterm neonates have dehydration rates as high as 40%; moreover, the smallest and most premature infants are generally placed in incubators, which may justify the higher dehydration rate observed (if accurately presented). Similar inconsistencies arise in the assessment every four hours: line 268 mentions dehydration risk 1.6 times higher when assessment every four hours was not performed, whereas the abstract suggests that assessment every four hours was protective (OR = 0.31).

5. In the limitations section, it is worth reinforcing the potential for bias. Even if the dehydration definition is deemed adequate, using 20% weight loss in the first week as a threshold appears too high. The usual range of weight loss in preterm infants is 10–15%; >15% is generally significant, making the dehydration threshold too specific and potentially underestimating its true incidence. Other authors (e.g., the article cited as Reference 22) have used >12% as the dehydration cutoff.

Moreover, nutritional intake and weight gain were not assessed. Inadequate weight gain can result from malnutrition (or sepsis/infection), yielding apparent dehydration due to low intake and complicating the differentiation between dehydration and malnutrition. Another consideration is the timing of admission (average of four hours). Infants admitted later may present more dehydrated, and comparing neonates admitted within the first hour versus those admitted later could justify a portion of the dehydration outcomes observed.

6. The discussion on dehydration associated with neonatal asphyxia is superficial and lacks a cited reference. The authors mention that incubators used for asphyxiated neonates may cause dehydration, but this contradicts the common clinical picture of SIADH, which is associated with fluid retention rather than dehydration. This point should be clarified.

Similarly, the role of maternal hypertension as a risk factor for dehydration is insufficiently explained and lacks supporting references. The latest meta analysis of maternal risk factors for dehydration (DOI:10.34763/jmotherandchild.20232701.d-24-00007) does not include maternal hypertension per se, suggesting that its role may be related to prematurity and its associated complications, rather than an independent risk.

7. Statistical inconsistencies should be revisited. In the tables, certain results (e.g., dehydration in neonates <1,500 g versus 1,500–2,500 g) appear inconsistent or unexpected. According to the data, 51 of 96 neonates weighing <1,500 g developed dehydration (around 53%), yielding a nonsignificant result. Meanwhile, 142 of 238 neonates weighing 1,500–2,500 g (around 60%) reportedly have an OR of 1.38 (0.48–3.98), p = 0.56. This inconsistency warrants rechecking for errors.

8. According to the STROBE checklist, the article generally satisfies the requirements for an observational cohort study, with a robust methodological design, appropriate statistical analyses, and well contextualized discussion. Minor improvements could be made in terms of conflict of interest declarations and statements of funding.

9. The reference list requires thorough review for accuracy and duplication. I noticed redundancies (e.g., References 2 and 17, and 6 and 18 are identical). Additionally, some formatting inconsistencies must be corrected (e.g., the term “request PDF” on line 395).

10. Finally, I recommend submitting the manuscript for review by an academic or scientific language editor. While the text is generally intelligible, a review for formal academic English (e.g., replacing “person days” with “patient days”) would further improve its quality and readability. Services such as the American Manuscript Editors or similar platforms can be helpful.

Thank you for submitting this important contribution. I am available for any further questions or clarification.

Reviewer #2: Authors have highlighted common causes of dehydration in developing countries except preterm with surgical conditions presenting with dehydration, which is also a major cause of poor outcome in preterm.

**Do you want your identity to be public for this peer review?** For information about this choice, including consent withdrawal, please see our Privacy Policy

Reviewer #1: **Yes: ** Marcos Alves Pavione

Reviewer #2: No

---

## [Author Response · Author response to Decision Letter 1]

24 Jul 2025

response for reviewers

Title: Incidence and predictors of dehydration among preterm neonates admitted in the neonatal intensive care unit of Southern Ethiopia hospitals: a prospective follow-up study

Author’s response to reviews: Point-by-point response to reviewers

Dear Editors/Reviewers,

Thank you, respectful reviewers and editors, for giving us a chance to revise our manuscript entitled “Incidence and predictors of dehydration among preterm neonates admitted in the neonatal intensive care unit of Southern Ethiopia hospitals: a prospective follow-up study.”

We addressed all the comments that rose from the reviewer and editor. The comments of the reviewers were insightful and important to improve the quality of our manuscript. Based on the reviewers’ comments, we have made detailed revisions and written responses for each reviewer’s comment. Finally, the clean version of the revised manuscript and track changes is uploaded. Please also note that page and line numbers given in the response letter refer to revised versions (not the original) of the manuscript.

Thank you.

Sincerely!

Each point-by-point comment with its respective response was presented as follows:

Reviewer reports:

Editors’ comment

Comment 1: Please ensure that your manuscript meets PLOS ONE's style requirements, including those for file naming.

Author's Response: Thank you, dear editors, for your suggestion. The authors ensure this manuscript was prepared according to the PLOS ONE journal style requirements. For further information, please check the track changes.

Comment 2: In the online submission form, you indicated that the data underlying the results presented in the study are available from the corresponding author upon reasonable request.

This policy applies to all data except where public deposition would breach compliance with the protocol approved by your research ethics board. If your data cannot be made publicly available for ethical or legal reasons (e.g., public availability would compromise patient privacy), please explain your reasons on resubmission, and your exemption request will be escalated for approval.

Author's Response: Thank you very much for your recommendation. We included all the necessary data in the manuscript, and we amended it in the online submission, as all data are fully available.

Comment 3: When completing the data availability statement of the submission form, you indicated that you will make your data available on acceptance. We strongly recommend all authors decide on a data sharing plan before acceptance, as the process can be lengthy and hold up publication timelines. Please note that, though access restrictions are acceptable now, your entire data will need to be made freely accessible if your manuscript is accepted for publication. This policy applies to all data except where public deposition would breach compliance with the protocol approved by your research ethics board. If you are unable to adhere to our open data policy, please kindly revise your statement to explain your reasoning, and we will seek the editor's input on an exemption. Please be assured that, once you have provided your new statement, the assessment of your exemption will not hold up the peer review process.

Authors’ response: Thank you for your recommendation. We describe it accordingly, as all relevant data are found within the manuscript and freely accessible in the Github. You can access in this link https://github.com/M12-21/DHN-IN-PRETERM-NEONATES.git.

Comment 4: Please amend either the title on the online submission form (via Edit Submission) or the title in the manuscript so that they are identical.

Author's Response: Appreciation! The authors amend accordingly. Please look at the title on the online submission and the manuscript.

Comment 5: Please amend either the abstract on the online submission form (via Edit Submission) or the abstract in the manuscript so that they are identical.

Author's Response: Thank you for your suggestion. We amend it accordingly.

Reviewer comments

Reviewer 1 Comments

First, I would like to commend the authors for selecting the topic of dehydration in preterm neonates admitted to a NICU. This is an important subject due to its high frequency and association with significant morbidity and mortality, which are even more pronounced in developing countries. The topic is sparsely addressed in the literature and clearly required considerable effort from the research team in both data collection and interpretation.

I have read the manuscript thoroughly and identified certain inconsistencies and points of uncertainty, which I have listed below:

Comment 1: A critical concern relates to the definition of dehydration used for data collection, as it serves as the foundation for the entire statistical analysis. The methodology does not clearly define “dehydration” in a detailed manner. It appears from the text that dehydration was defined as a 20% loss of birth weight in the first week and a 10% loss after the first week, but other clinical signs and criteria associated with dehydration are mentioned, making it unclear how these were used in conjunction with weight loss. It would be ideal if the authors clearly stated the definition, for example, “dehydration was defined as a weight loss combined with at least one of the following clinical signs,” or “dehydration was defined as weight loss and all clinical signs listed,” or “dehydration was defined as weight loss and two or more clinical signs,” etc. Additionally, I would like to review the data collection form used, as this would help clarify the approach. This point needs to be rewritten to eliminate ambiguity, as it critically affects the rest of the analysis.

Author's Response: Good insight! Thank you very much for your valuable comment. We accept and clearly describe the definition of dehydration. The authors define dehydration as preterm neonates having weight loss (≥20%) in the first week of life (current weight minus birth weight) and ≥10% after the first week of life) combined with two or more of the following: Signs of oral dry mucosa, depressed fontanel, feeble pulse, increased serum sodium (Na>145 mEq/L), increased urine specific gravity (>1.020), urine osmolality (>400 mosm/L), or decreased urine output (<1 ml/kg/hr) were considered as dehydration. Please check the track changes in lines 158 to 165.

Comment 2: I recommend including a reference for the definition of dehydration in preterm infants, if available. The text should state clearly whether the definition was based on a prior published article, another author’s experience (with a cited source), or created exclusively for this study.

Authors’ response: Thank you again for your constructive comment. As per researchers’ efforts and knowledge, we couldn’t find available evidence that clearly defines dehydration for preterm neonates. But we used ideas from physicians’s experience (one of the authors of this study is a pediatrician, Dr. Mitsiwa) and the case definitions of studies conducted in Kenya, the Netherlands, and California and created our own definitions to make it all-inclusive and reduce bias. We kindly request to see the track change in submitted manuscript line 165.

Comment 3: Another point for review is the measurement unit for dehydration. When comparing rates across studies, it is crucial to ensure that comparable units are used. In the literature I reviewed, including articles cited by the authors’ the incidence of dehydration is highly variable due to differing definitions and objectives (e.g., dehydration in exclusively breastfed infants versus dehydration due to excessive fluid loss). Units of measurement also vary (percentage of dehydrated patients, dehydrated patients per 1,000 live births, dehydrated patients per 1,000 exclusively breastfed neonates, etc.). I could not find any publication using the same measurement (dehydration per 1,000 patient days) as used in this study. This discrepancy complicates direct comparisons with existing data.

Author’s response: we appreciate it! The authors accept the comment. As you mentioned, the unit of measurement and the population in studies conducted in the Netherlands and California differ from the unit of measurement of incidence of dehydration in this study. Even though the authors use these study findings as a comparison, we discuss the variation in lines 284 to 287 and put it as a limitation in the limitation section, lines 353 to 354, since there were no equivalent studies conducted regarding dehydration among preterm neonates with similar measurements. We appreciate if you suggest any possibilities.

Comment 4: Inconsistencies arise in the text when the authors compare their results with the literature using differing units. For example, lines 238–239 seem to imply that an incidence of 41 per 1,000 patient days is equivalent to 41%, which is incorrect.

Author’s response: Thank you very much for your comment. We try to check lines 238-239 in the submitted manuscript. This sentence concept does not mean that the incidence of 41 per 1,000 patient days is equivalent to the percentage (41%). But the intent of this description is the percentage of new cases, which is the percentage of preterm neonates who develop dehydration among study participants, i.e., 148 per 363. When we divide it, we find approximately 41%. So, it indicates the percentage of preterm neonates who develop dehydration during the follow-up among 363 preterm neonates admitted to the NICU. However, an incidence of 41 per 1,000 patient days indicates the incidence rate of preterm dehydration from a total follow-up time of 3,546 person-days of observations.

Comment 5: More generally, when the authors cite literature about dehydration definitions, two key studies are mentioned, yet reference 11 is not cited in the section dealing with definitions and comparisons. Upon review, I found that:

• In the Netherlands study (Reference 11), dehydration was defined as weight loss >10% of birth weight combined with a serum sodium ≥150 mEq/L. To minimize bias related to low intake and weight loss due to infection, neonates with UTIs and GERD were excluded.

• In the California study (Reference 22), dehydration was defined as either (a) a serum sodium ≥150 mEq/L and/or (b) weight loss ≥12%.

Authors’ response: Thank you for your suggestion. We accept your comment and cite the references in the section dealing with definition and comparison. The authors ’kindly request to look at it in the track change.

Comment 6: In lines 281–282, I could not locate the cited California reference (22), and the Netherlands reference is not mentioned (although I assume it is Reference 11). The Dutch article uses dehydration incidence per 100,000 exclusively breastfed neonates, while the Californian reference uses an incidence of 2.1 per 1,000 live births. These differing units must be clearly explained in the text, making it evident that the objectives and patient populations differ from those of the present study (dehydration in preterm neonates in a NICU).

Author’s response: Acknowledged! We accept it and made a correction; please refer to the tracked change in lines 284-87.

Comment 7: It would be helpful to compare the current results with those from other studies using the same metric (dehydration per 1,000 patient days).

Author’s response: Thank you for the suggestion. As you state, it is better to compare the study findings from studies with similar metrics. However, to the researchers’ knowledge, no available studies have been conducted directly on dehydration of preterm neonates. Due to this, we couldn’t find studies with similar metrics. Consequently, the authors used related studies conducted in neonates.

Comment 8: The results section contains inconsistencies between data presented in tables and those stated in the conclusion. For example, on line 266, an odds ratio (OR) of 3.38 suggests that incubators are a risk factor for dehydration. However, in the abstract, an OR of 0.19 is stated, implying a protective effect. This discrepancy must be resolved.

Authors’ response: Thank you for your valuable comment. Sorry for this typing error. We accept and amend accordingly. Please refer to lines 39-40.

Comment 9: Moreover, upon closer inspection of the tables, it is unclear how bed types were classified. The text mentions incubators, radiant warmers, common cribs, and phototherapy units. However, it is ambiguous whether “common cribs” means a regular hospital crib or a bed adjacent to the mother. Additionally, why was phototherapy treated as a bed category, given that it can be administered in any of these environments? This may introduce confounding, as dehydration attributed to an incubator might, in fact, relate to phototherapy exposure.

Author’s response: Thank you very much for your comment. We classify the beds based on the neonates kept during the follow-up time. That means there were neonates kept in regular hospital beds (beds without radiant warmers, incubators, and phototherapy); neonates were also kept in beds with radiant warmers, incubators, and phototherapy. So, neonates kept in regular beds are considered as neonates kept in regular beds, neonates kept in beds with radiant warmers are considered as neonates kept in radiant warmers, neonates kept in beds with incubators are considered as neonates kept in incubators, and neonates kept in beds under phototherapy are considered as neonates kept under phototherapy during follow-up time. We use this classification to appreciate the difference in terms of risk for dehydration among neonates kept in bed, in an incubator, under a radiant warmer, and under phototherapy. To make it clear for readers, the authors amend the wordings of these terms in table 4.

Comment 10: Importantly, the results and tables do not assess gestational age as a potential risk factor for dehydration. According to literature, extremely preterm neonates have dehydration rates as high as 40%; moreover, the smallest and most premature infants are generally placed in incubators, which may justify the higher dehydration rate observed (if accurately presented).

Authors’ response: a million thanks for your valuable comment. As you stated in different literatures, extreme preterm neonates were at risk for dehydration. By considering this, we also include the gestational age as one variable in the bivariable analysis. But its p-value becomes greater than 0.25 (p-value=0.39), which is used as a cut point to transfer variables after bivariable analysis to multivariable analysis in this study. Even if it didn’t fulfill the criteria to transfer to multivariable analysis, we try to involve it also in multivariable analysis by considering its clinical importance. However, while we tested model fitness, it also broke the model assumptions. Due to this, we didn’t include the gestational age in the multivariable analysis.

Comment 11: Similar inconsistencies arise in the assessment every four hours: line 268 mentions dehydration risk 1.6 times higher when assessment every four hours was not performed, whereas the abstract suggests that assessment every four hours was protective (OR = 0.31).

Authors’ response: We appreciate it! Sorry for this typing error. We accept and amend it. Please refer to lines 40-42 in the track change.

Comment 12: In the limitations section, it is worth reinforcing the potential for bias. Even if the dehydration definition is deemed adequate, using 20% weight loss in the first week as a threshold appears too high. The usual range of weight loss in preterm infants is 10–15%; >15% is generally significant, making the dehydration threshold too specific and potentially underestimating its true incidence. Other authors (e.g., the article cited as Reference 22) have used >12% as the dehydration cutoff.

Authors’ response: The authors strongly acknowledged the reviewer for giving this important comment. As you mentioned, the usual weight loss in neonates was 10-15%. But using 15% as a cut point for both preterm neonates below 1

---

## [Decision Letter · Decision Letter 1]

12 Aug 2025

Dear Dr. Ayehu,

We look forward to receiving your revised manuscript.

Kind regards,

Ricardo Q. Gurgel, PhD

Academic Editor

PLOS ONE

Journal Requirements:

Reviewers' comments:

Reviewer's Responses to Questions

**Comments to the Author**

Reviewer #1: All comments have been addressed

Reviewer #2: All comments have been addressed

2. Is the manuscript technically sound, and do the data support the conclusions?

Reviewer #1: Yes

Reviewer #2: Yes

3. Has the statistical analysis been performed appropriately and rigorously?

Reviewer #1: Yes

Reviewer #2: Yes

4. Have the authors made all data underlying the findings in their manuscript fully available?

Reviewer #1: Yes

Reviewer #2: Yes

5. Is the manuscript presented in an intelligible fashion and written in standard English?

Reviewer #1: Yes

Reviewer #2: (No Response)

Reviewer #1: First, I would like to congratulate the authors for their effort in implementing the suggested revisions in a timely manner and for their maturity in recognizing that the critiques were constructive, aiming to make the manuscript more scientifically robust.

Following the revisions, the main points were adequately addressed, and there is a substantial improvement in both the clarity of the text and the comprehensibility of the information. At this stage, only a few minor corrections remain, mostly related to writing style, as outlined below:

1. Although marked as addressed, the term person-days was not replaced by patient-days, which is the correct term for this statistical measure. In some parts of the manuscript, person-days remains, whereas in others it was replaced with preterm-days, which is not used in the literature (e.g., lines 37 and 241). I suggest standardizing to patient-days throughout the text.

2. In line 38, parentheses appear duplicated.

3. Two abbreviations appear in the text without prior full spelling: TBW (line 71) and ANC (line 220).

4. The term mensuration should be replaced with menstrual in line 169.

5. To maintain uniformity of reporting, the percentage of hypothermic cases should be included (lines 228–229).

6. In the limitations section, it is important to add the justification for not using gestational age, as explained in the response letter to the editor.

7. Finally, in the section on justifications for dehydration in the context of neonatal asphyxia, the rationale still does not adequately align with the physiology of dehydration in asphyxia.

Below, I provide a review text that may be used to strengthen the explanation:

Justification for Increased Dehydration in Neonates with Neonatal Asphyxia

The increased dehydration observed in neonates with neonatal asphyxia can be explained by the pathophysiology of the hypoxic–ischemic insult and its renal, hemodynamic, and metabolic repercussions.

1. Redistribution of Blood Flow (“Centralization”)

During asphyxia, there is intense sympathetic activation and a rise in catecholamines. Cardiac output is redistributed to vital organs (brain, heart, adrenal glands), with peripheral vasoconstriction and reduced renal and splanchnic blood flow. This renal hypoperfusion impairs the ability to concentrate urine and retain sodium and water, contributing to increased urinary losses during the post-reperfusion phase.

2. Hypoxic–Ischemic Renal Injury

Asphyxia may cause acute tubular injury (acute tubular necrosis) and nephron dysfunction. Typical alterations include urinary loss of sodium (natriuresis) and chloride, in addition to impaired tubular water reabsorption. This leads to osmotic diuresis and hyponatremia, increasing fluid deficit.

3. Hormonal Alterations

Hypoxia reduces both secretion and responsiveness to vasopressin (ADH) and to the renin–angiotensin–aldosterone system in the initial phase. Tubular resistance to ADH promotes free water loss, while reduced aldosterone activity increases sodium loss, further aggravating dehydration.

4. Increased Metabolic Demand After Reperfusion

Following resuscitation, a hypermetabolic state develops due to systemic inflammatory response and tissue repair. This increases heat production, oxygen consumption, and fluid requirements, accelerating the onset of a negative fluid balance.

5. Additional Losses

Newborns with asphyxia often require mechanical ventilation (sometimes with suboptimal heating and humidification) or phototherapy, both of which increase insensible water loss. Metabolic acidosis may also promote compensatory diuresis.

Pathophysiological Summary

Neonatal asphyxia induces blood flow centralization and hypoxic–ischemic tubular injury, leading to natriuresis and impaired urine concentration. Combined with hormonal dysregulation and increased insensible losses, these changes predispose affected neonates to more rapid and severe dehydration compared to those without asphyxia.

References

1. Selewski DT, Charlton JR, Jetton JG, Guillet R, Mhanna MJ, Askenazi DJ, Kent AL. Neonatal acute kidney injury. Pediatrics. 2015;136(2):e463–e473.

2. Jetton JG, Askenazi DJ. Update on acute kidney injury in the neonate. Curr Opin Pediatr. 2012;24(2):191–196.

3. Perlman JM, Tack ED, Martin T, Shackelford G, Amon E. Acute systemic organ injury in term infants after asphyxia. Am J Dis Child. 1989;143(5):617–620.

4. Kaur S, Jain S, Saha A, Chawla D, Parmar VR, Basu S. Incidence and predictors of acute kidney injury in term neonates with perinatal asphyxia. J Clin Neonatol. 2017;6(4):231–235.

5. Guignard JP, Drukker A. Why do newborn infants have a high plasma creatinine? Pediatrics. 1999;103(4):e49.

6. World Health Organization. Neonatal emergency protocols. Geneva: WHO; 2023.

I remain at your disposal for any further clarification.

Reviewer #2: (No Response)

**Do you want your identity to be public for this peer review?** For information about this choice, including consent withdrawal, please see our Privacy Policy

Reviewer #1: **Yes: ** Marcos Alves Pavione

Reviewer #2: **Yes: ** Samuel Wabada, Department of Surgery, division of Paediatric Surgery, University of Maiduguri, Borno State, Nigeria

---

## [Author Response · Author response to Decision Letter 2]

5 Sep 2025

Title: Incidence and predictors of dehydration among preterm neonates admitted in the neonatal intensive care unit of Southern Ethiopia hospitals: a prospective follow-up study

Author’s response to reviews: Point-by-point response to reviewers

Dear Editors/Reviewers,

Thank you, respectful reviewers and editors, for giving us a chance for the second round to revise our manuscript entitled “Incidence and predictors of dehydration among preterm neonates admitted in the neonatal intensive care unit of Southern Ethiopia hospitals: a prospective follow-up study.”

We addressed all the comments that arose from the reviewer and editor. The comments of the reviewers were insightful and important to improve the quality of our manuscript. Based on the reviewers’ comments, we have made detailed revisions and written responses for each reviewer’s comment. Finally, the clean version of the revised manuscript and track changes are uploaded separately. Please also note that page and line numbers given in the response letter refer to revised versions (not the original) of the manuscript.

Thank you.

Sincerely!

Each point-by-point comment with its respective response was presented as follows:

Reviewer comment

Comment 1: Although marked as addressed, the term person-days was not replaced by patient-days, which is the correct term for this statistical measure. In some parts of the manuscript, person-days remain, whereas in others it was replaced with preterm-days, which are not used in the literature (e.g., lines 37 and 241). I suggest standardizing to patient-days throughout the text.

Author's response: Great apologies!!! We accept and modify it accordingly throughout the page.

Comment 2: In line 38, parentheses appear duplicated.

Authors' response: Thank you for your suggestion: the authors accept and correct it. Please check line 38 in the track change.

Comment 3: Two abbreviations appear in the text without prior full spelling: TBW (line 71) and ANC (line 220).

Authors' response: a million thanks! The authors made the abbreviations full spelling. Please see lines 71 and 220

Comment 4: The term mensuration should be replaced with menstrual in line 169.

Authors' response: great appreciation!!! We modified according to your suggestion. Please look at line 169

Comment 5: To maintain uniformity of reporting, the percentage of hypothermic cases should be included (lines 228–229).

Authors' response: Thank you very much for your constructive comment!!! We put the percentage as your suggestion. Please refer to line 229.

Comment 6: In the limitations section, it is important to add the justification for not using gestational age, as explained in the response letter to the editor.

Authors' response: Thank you for the suggestion. The authors state it in the limitation section of the manuscript, as your recommendation. Please refer to lines 359-65.

Comment 7: Finally, in the section on justifications for dehydration in the context of neonatal asphyxia, the rationale still does not adequately align with the physiology of dehydration in asphyxia.

Authors' response: Thank you very much for your suggestion and support. The authors accept your suggestion and have modified it by incorporating the points that you suggested. Please check lines 327-37.

---

## [Editor Report · Decision Letter 2]

9 Sep 2025

Incidence and predictors of dehydration among preterm neonates admitted to neonatal intensive care unit of Southern Ethiopia hospitals: a Prospective follow-up study

PONE-D-25-27567R2

Dear Dr. Ayehu

We’re pleased to inform you that your manuscript has been judged scientifically suitable for publication and will be formally accepted for publication once it meets all outstanding technical requirements.

Kind regards,

Ricardo Q. Gurgel, PhD

Academic Editor

PLOS ONE
---

## [Editor Report · Acceptance letter]

PONE-D-25-27567R2

PLOS ONE

Dear Dr. Akele,

I'm pleased to inform you that your manuscript has been deemed suitable for publication in PLOS ONE. Congratulations! Your manuscript is now being handed over to our production team.

Kind regards,

on behalf of

Professor Ricardo Q. Gurgel

Academic Editor

PLOS ONE